# Multimorbidity and health-related quality of life among patients attending chronic outpatient medical care in Bahir Dar, Northwest Ethiopia: The application of partial proportional odds model

**Fantu Abebe Eyowas**[1]*, **Marguerite Schneider**[2], **Shitaye Alemu Balcha**[3], **Sanghamitra Pati**[4], **Fentie Ambaw Getahun**[1]

**1** School of Public Health, College of Medicine and Health Sciences, Bahir Dar University, Bahir Dar, Ethiopia, **2** Alan J Flisher Centre for Public Mental Health, Department of Psychiatry and Mental Health University of Cape Town, Cape Town, South Africa, **3** School of Medicine, College of Medicine and Health Sciences, University of Gondar, Gondar, Ethiopia, **4** ICMR-Regional Medical Research Center, Bhubaneswar, India

* fantuabebe@gmail.com

## Abstract

### Background

Multimorbidity, the presence of two or more chronic non-communicable diseases (NCDs) in a given person affects all aspects of people's lives. Poor quality of life (QoL) is one of the major consequences of living with multimorbidity. Although healthcare should support multi-morbid individuals to achieve a better quality of life, little is known about the effect of multi-morbidity on the QoL of patients living with chronic conditions. This study aimed to determine the influence of multimorbidity on QoL among clients attending chronic outpatient medical care in Bahir Dar, Northwest Ethiopia.

### Methodology

A multi-centered facility-based study was conducted among 1440 participants aged 40+ years. Two complementary methods were employed to collect sociodemographic and disease related data. We used the short form (SF-12 V2) instrument to measure quality of life (QoL). The data were analyzed by STATA V.16, and a multivariate partial proportional odds model was fitted to identify covariates associated with quality of life. Statistical significance was considered at p-value <0.05.

### Principal findings

Multimorbidity was identified in 54.8% (95% CI = 52.2%-57.4%) of the sample. A significant proportion (33.5%) of the study participants had poor QoL and a quarter (25.8%) of them had moderate QoL. Advanced age, obesity and living with multimorbidity were the factors

**Data Availability Statement:** All data are in the manuscript and/or Supporting information files.

**Funding:** This work was partially funded by Bahir Dar University [grant number: RCS/003/21]. The funders had no role in study design, data collection and analysis, decision to publish, or preparation of the manuscript.

**Competing interests:** The authors have declared that no competing interests exist.

associated with poor QoL. Conversely, perceived social support and satisfaction with care were the variables positively associated with better QoL.

## Conclusion

The magnitude of multimorbidity in this study was high and individuals living with multimorbidity had a relatively poorer QoL than those without multimorbidity. Care of people with chronic multiple conditions has to be oriented to the realities of multimorbidity burden and its implication on QoL. It is also imperative to replicate the methods we employed to measure and analyze QoL data in this study for facilitating comparison and further development of the approaches.

## Introduction

The increasing demographic and social changes with ageing populations are leading to rapid epidemiological transitions, including the rise of chronic non-communicable diseases (NCDs) and multimorbidity [1].

Multimorbidity refers to the presence of two or more coexisting long-term conditions, being related or not in a given person [2]. Despite the inconsistency in the methodologies employed to define and measure multimorbidity [3], the burden of multimorbidity is shown to be growing globally [4, 5], in LMICs [6, 7] and Ethiopia [8, 9]. The prevalence of multimorbidity is projected to double by 2035, and the majority of the people surviving beyond 65 years will have four or more chronic conditions [10, 11].

Advanced age [4, 12–15], socioeconomic deprivation [15], obesity [16], female sex [17, 18], physical inactivity [19], use of tobacco and alcohol [20, 21] and psychosocial factors such as poor social networks and external locus of control [22, 23] were the most common factors associated with multimorbidity in the global literature.

Multimorbidity affects all aspects of patients' lives. Poor quality of life (QoL) along with disability, functional decline and high health care costs are major consequences of living with multimorbidity [19, 24, 25]. The effect of multimorbidity on QoL was reported to be high among middle-aged, elderly and female population groups, and individuals with comorbid mental illnesses [17].

Living with multimorbidity is beyond the mere sum of individual chronic conditions [26]. The specific disease clusters that an individual is living with would have a different effect on their physical and psychological functioning [27]. For instance, the health-related QoL has always been lower among people living with multimorbidity compared to those without multimorbidity [28].

Lifelong presence of multimorbidity is also posing a significant challenge for the health system [29]. Individuals living with different types and combinations of NCDs may have different needs and priorities [30]. Nevertheless, too little attention is paid to what matters to people living with multiple health problems [31, 32].

Moreover, the current model of care and guidelines being developed at a time when single disease frameworks were predominant, tend to focus on diseases in isolation rather than the needs and circumstances of the person with complexity care needs as a whole [33, 34]. As a result, people living with multiple conditions will be in contact with multiple health professionals who may not communicate each other and with inadequate data flow across the

healthcare system, leading to a fragmented, uncoordinated and siloed patient management [2, 35]. Further, the rapid emergence of infections such as COVID-19 fuels the complexity and adds a huge burden to the health system with worsening outcomes for patients with preexisting chronic diseases and multimorbidity [36, 37].

Several valuable studies investigated the relationship between QoL and multimorbidity [38, 39]. However, most of them were conducted in high-income countries and the tools employed to measure QoL among people living with multimorbidity have not been consistent [40]. Some studies in high-income countries used Euro QoL (EQ-5D-5l) [41, 42], while others used either WHOQOL brief [40] or SF-36 [40, 43] or SF-12 tools [40, 44–46]. Although the use of all of these tools has been widespread, the short form (SF-12) version is an efficient algorism to reproduce SF-36 tool to measure health related quality of life [47].

The observed variations in the existing literature also included the methods of data analyses used [48]. The way the data have been generated is particularly important for analyzing quality of life assessments scores [49]. Health related QoL is often measured by Likert-type scales and the scores are treated as if they were continuous and normally distributed, which often is not the case [50]. Scholars in the field have noted that analyzing ordinal data as if they were a metric one (continuous) can systematically lead to distorted effect-size estimates, inflated errors rates and inaccurate parameter estimates [51, 52].

Neither are the methods used for binary data adequate to fully take account of the properties of ordered outcomes such as QoL [48, 53]. Hence, a more sensitive and comprehensive model is required. Evidence suggests that the ordinal regression models are superior to the methods commonly used to analyze data of an ordered nature [54, 55]. The ordinal models provide better theoretical interpretation and numerical inference than the metric (linear) models for ordered outcomes [56, 57].

The ordinal regression model provides unbiased estimates when the data meet the proportional odds (PPO) assumption [53, 56]. The PPO assumption implies that all observations have a common variance on the underlying continuum, and the coefficients that describe the relationship between, say, the lowest versus all higher categories of the response variable are the same except in the cut-off points [49, 55, 58]

However, it is often difficult to find data for which a proportional odds model is a plausible description, and evidence suggests that the assumptions of the ordered logit (proportional odds) model are frequently violated [54]. When the given data violates the parallel regression assumption, a more realistic approach, the partial proportional odds (PPO) model would be suitable [54]. This model is robust in revealing unobserved heterogeneity in the group and identify correlates contributing to negative health outcomes, including impaired QoL [48, 53]. The primary reason for the formulation of the partial proportional odds models is to relax the stringent assumption of constant odds ratio over all the cut-points for a given covariate [55].

Supporting people living with long-term conditions to maintain a good quality of life is one of the key challenges facing the healthcare and social care systems today [25]. Studies have suggested that the management of patients with multimorbidity should take into account the impact of multimorbidity on a person's quality of life and the person's own priorities [59, 60]. However, little is known about the effect of multimorbidity on health-related QoL in Ethiopia. If health systems are to meet the needs and priorities of individuals living with multimorbidity, we need to adequately measure the magnitude and impact of multimorbidity on QoL among patient population with chronic conditions.

The present study aimed to understand the influence of multimorbidity on QoL among individuals attending chronic outpatient medical care in Bahir Dar, Northwest Ethiopia.

## Materials and methods

This facility-based study was conducted in eight health facilities providing chronic NCDs care in Bahir Dar City, Ethiopia. This study presents the result of one part of an ongoing research project in the region. All methods were carried out in accordance with relevant guidelines and regulations. The detail of the methods employed in this study has been published elsewhere [61].

### Design

This multi-center, facility-based cross-sectional study was conducted in public and private health facilities rendering health services in Bahir Dar City, Ethiopia. The city is the capital of the Amhara regional state, the second most populous region in the country with a population of about 31 million.

### Study setting and population

This study was conducted in five hospitals (three public and two private) and three private higher/specialty clinics in the city. These facilities also serve as referral center for primary care facilities surrounding the regional capital. Chronic NCDs care and management in these facilities is supposed to be provided in a relatively uniform fashion using the national NCDs treatment guideline [62]. However, the nature of patients visiting these facilities may vary and there remains a concern on the quality and affordability of NCDs care in public hospitals and private health facilities, respectively.

Only facilities which were providing chronic NCDs care by medical doctors (general practitioners or specialist physicians) for at least a duration of one year prior to the data collection period were considered. Older adults (40 years or more) diagnosed with at least one NCD and on chronic diseases follow up care for at least six months at the time of data collection were recruited into the study. However, pregnant women and individuals who were too ill to be interviewed and admitted patients were excluded.

### Sample size

Key issues considered to estimate the sample size required were the nature of the dependent and predictor variables and the anticipated data analysis techniques. The input values: $\alpha$ (type I error = 0.05), power (1-$\beta$ = 90), confidence level (95%) and the estimated non-response and attrition during follow-up (20%) remain constant. The sample size yielded by the general linear multivariate model with Gaussian errors (GLIMMPSE) sample size and power calculator (32–34) formula was chosen owing to its adequacy to answer all the study objectives compared to other techniques. Based on the given assumptions and the approach we used, the calculated sample size required became 600. As the nature of participants is likely to be different by the type of facility (public or private) where they receive care, we employed stratification to ensure fair representation in the sample for important sub-groups analysis. Hence, a design effect of 2 was considered to avoid the possible loss of sample during stratification. Adding 20% to the possible loss to follow-up (considering the longitudinal study) and nonresponse, the sample size needed was calculated to be 1440. About half (n = 728, 50.84%) of the participants were enrolled from public facilities.

### Sampling technique

A stratified random sampling method was employed for recruiting eight eligible facilities and a corresponding number of participants. The sample size from each facility was determined

based on the notion of probability proportional to size (PPS) using the pool of chronic NCD patients ($\geq$ 40yrs) registered for chronic follow-up over the year preceding our assessment (January—December 2020) in each participating facility. Health facilities and eligible clients were randomly selected for the study.

## Definition and measurement of dependent variable (HRQoL)

HRQoL (stated as QoL in this study) is defined as individuals' perception of their position in life in the context of physical, psychological and social functioning and well-being [63]. QoL was measured using the interviewer-administered short form (SF-12 V2) assessment tool [64, 65], which is derived from the SF-36 QoL assessment tool [47].

The SF-12 tool is extensively validated and widely used generic tool for measuring QoL in multimorbidity across different contexts, including Sub-Saharan Africa [46, 66, 67]. The tool was translated and pilot tested according to the study protocol we published [61]. The tool measures eight health aspects, namely physical functioning (PF), role limitations due to physical health problems (RP), bodily pain (BP), general health perceptions (GH), vitality (VT), social functioning (SF), role limitations due to emotional problems (RE), and mental health (psychological distress and psychological well-being) (MH). Two summary measures are derived from the SF-12: physical health (Physical Component Summary-PCS) and mental health (Mental Component Summary -MCS). However, owing to the possibility of correlation (lack of uni-dimensionality) between the PCS and MCS scores, some studies criticized the use of these scoring algorithms and recommended raw sum scores instead [47, 52]. The use of a single raw sum score enables a consistent assessment of the impact of multimorbidity and how this varies across a given population [68]. Thus, we applied this approach for analyzing the QoL data.

First, we reverse coded the scores for items 1, 9 and 10 and computed the raw total. The overall scores were scaled from 0 to 100, with 0 representing worst health [69]. Although popularly used in previous studies, the notion of fitting linear regression models to summarize categorical data such as the QoL data has been questioned [49, 52]. The linear regression models may potentially lose important variability in the data particularly when the QoL data is collected by Liker-type scales such as the SF-12 tool [48, 53]. Recent advances in the field recommend the interpretation of QoL rather as a categorical (group continuous) variable than as a metric variable [49]. Studies suggest that ordinal regression models (OLR) are superior to other method for analyzing ordinal data, including health-related QoL data [49, 50]. Hence, we ranked the scaled QoL scores into three ordered and non-overlapping categories as poor QoL (a scaled value <75), moderate QoL (scaled value from 75–89.9) and high QoL (scaled value from 90–100) [53]and fitted into the OLR and partial proportional odds (PPO) models.

## Measurement of independent variables

Independent variables including socio-demographic characteristics [age, gender, education, marital status, residence and occupation] were assessed using validated tools. Data to calculate body mass index (BMI) and waist to hip circumference were directly measured from patients according to the approaches described in our study protocol published elsewhere [61].

Social networking and support systems were assessed through face-to-face interview using the Oslo Scale [70]. The tool was translated and pilot tested among 29 patients attending chronic care in health facilities which were not selected for the actual study. A scale ranging from 3–8 was interpreted as poor social support, 9–11 moderate social support and 12–14 strong social support) [70].

A wealth Index at a household level was generated from a combination of material assets and housing characteristics [71]. The Wealth index was scored using principal component analysis (PCA) technique. The score was classified into quintiles, for urban and rural residents separately., Quintile 1 represents the poorest and quintile 5 the wealthiest [72]. It was collapsed into three classes as low, middle and high income to ease the analysis and interpretation.

Multimorbidity was operationalized as the co-occurrence of two or more of the chronic NCDs, including hypertension, diabetes, heart diseases (heart failure, angina and heart attack), stroke, bronchial asthma, chronic obstructive pulmonary diseases (COPD), depression, cancer, musculoskeletal disorders (arthritis, chronic back pain and osteoporosis), thyroid disorders (hyperthyroidism and multinodular goiter), chronic kidney disease, gastrointestinal disorders (chronic liver, gall bladder and gastric diseases) and Parkinson's disease (PD). The list of NCDs identified for the study was determined based on a review study [6] and preliminary and pilot studies conducted prior to the main study. Information on these chronic conditions was obtained from interview and review medical records using standardized tools [61].

Functional status (limitation) was assessed using the WHO'S Disability Assessment tool (WHODAS 2.0) [73]. The tool is a five-point scale (none = 1, mild = 2, moderate = 3, severe = 4, extreme/cannot do = 5). The results of score on the 12 items were summed up and categorized into three (with a score ≤50 as no limitation, 51–75 moderate limitation and 76–100 severe limitation). The 12 items WHODAS 2.0 has been validated and used in Ethiopia [74].

## Data collection tools and procedures

As mentioned above, the data were collected mainly from two different sources: interview and review of medical records. The combined questionnaire to collect the data was translated to Amharic (local language) and pilot tested for cross-cultural adaptability based on standard protocols [75, 76]. The data were collected by the Kobo Toolbox software [77]. Patients were interviewed and assessed following their regular consultation visit. Data were primarily collected by ten graduate nurses recruited from institutions outside the study facilities. Moreover, physicians and nurses working in the chronic care unit were facilitated in the data collection process.

To ensure good data quality, data collectors and supervisors were provided with a two-day training detailing the study, including obtaining written consent, conducting face-to-face interviews, performing physical measurement, medical record review and navigating through the questionnaires in the Kobo toolbox platform preloaded into their smart phones. The data collection process was monitored by trained supervisors, and the principal investigator. The data sent to the Kobo toolbox server were checked daily for completeness, accuracy and clarity. Feedback and coaching support were given for the data collectors on the quality and completeness of the information they submitted daily. Once the data were sent to the server, they were deleted from the data collection devices, and deletion of any data remained from smart phones and the questionnaire was made by the PI immediately after completion of the data collection.

## Data analysis

The data from the Kobo toolbox server were downloaded into excel spreadsheet and exported to SPSS V. 21 for cleaning and were analyzed using STATA V. 16. Descriptive statistics were computed to describe the sociodemographic characteristics of participants. The number of individual chronic conditions and multimorbidity status was determined by combining the data from patient interviews and medical record reviews.

In addition, the proportion of individuals falling into each of the QoL category was calculated. QoL as an ordered outcome was categorized as low, moderate and high, and coded as 0,

1 and 2, respectively while fitting into the ordinal logistic regression model. The association between each explanatory variable and QoL was assessed separately and model fitness was checked using the proportional odds (test for parallel regression) assumption [54]. The proportional odds (PO) assumption is said to be satisfied when we fail to reject the null hypothesis (a p-value of >0.05 in the Brant test) in the ordinal logistic regression model [54, 55].

For variables which fail to satisfy the PO assumptions, the partial proportional odds (PPO) model is more appropriate [48, 53], as the OLR model cannot fit the data well [55]. The partial proportional odds (PPO) model bridge the gap between ordered and non-ordered modeling frameworks [56, 78]. While the ordinal logistic regression model is restrictive and assumes that the effect of independent variables remain the same (fixed) for all levels of the dependent variables, the PPO allows the independent variables to take into account the individual differences in their effect on the dependent variables[55, 79]. Compared to the OLR model, the PPO has performed well in studies that compared different analytical models fitted for QoL data [49, 53]. Hence, we fitted the PPO (gologit2, autofit lrforce and gologit2, autofit lrforce gamma commands) model for determining covariates associated with QoL and to clearly identify the variables which violate the assumptions.

The independent variables fitted into the PPO model include residence, sex, age, marital status, education, BMI, social support, SES, multimorbidity, self-rated functional capacity and satisfaction with care. Independent variables having more than two categories were collapsed into two categories while fitting the PPO model [54]. The association between QoL and independent variables was assessed by fitting univariate and multivariate odds ratio (OR) with 95% confidence intervals and p-values are reported for each of the independent variable analyzed. Variables having a p-value <0.2 were fitted into multivariable PPO models to predict the adjusted effect of the independent variables on QoL. Before running the multivariable analysis, multi-collinearity between independent variables was checked using the Variance Inflation Factor (VIF) and variables were not strongly correlated (the highest value was 1.05). To make the interpretation more straightforward, we expressed the effects in terms of odds ratio rather than as regression coefficients [53]. In all cases, a p-value < 0.05 was taken as a statistically significant relationship.

We have conducted the univariate and multivariate analyses using the svyset, svy and gsvy prefixes to account for the potential difference of the sample drawn from the two strata (public and private health facilities) [80].

Looking at the Brant test of significance, most of the explanatory variables satisfied the proportional odds assumption. However, two independent variables (social network scale and satisfaction with care) violated the assumption of parallel lines regression (p-value ≤0.05), warranting the application of multivariate partial proportional odds model. Considering the effect of stratification, we also applied syv prefix before the ologit command, and output shows changes in the parameter estimates, brant test values and level of significance in the univariate analysis.

As stated above, the nature of the independent variables necessitated fitting of the partial proportional odds (PPO) model. The partial PPO model allows variables that meet the assumption to be modeled with the proportional odds assumption, whilst allowing others to have odds ratios that vary for the different categories that are compared. Only the variables with a p-value <0.2 in the univariate (using svy syntax) ordinal logistic regression analysis were fitted into the multivariate partial proportional odds model using gsvy (glogit2) command accounting for the stratified sampling.

Fitting the partial proportional odds assumption requires that the independent variables to have only two categories. Accordingly, except for age and social support score, we coded independent variables as a binary (0, 1) response category, where higher values were coded as "1"

and low values were given "0" and treated as a base category. Therefore, sex was coded as male (0) and female (1), SES as low (0) and middle or high (1), BMI as ≤24.99 (0) and ≥25 (1), multimorbidity was scored as no (0) and yes (1), functioning was scored as severe limitation (0) and no or mild limitation (1) and satisfaction was scored as not satisfied (0) and satisfied (1). Whereas, age and social support scales were treated as continuous independent variables.

Without adjustment and weighting, application of a stratified sampling method could affect parameter estimates of a given sample [56, 80]. Hence, we applied a more stringent model of analysis using the gsvy: gologit2 auto lrforce command.

In the final multivariate ordinal logistic regression model, the Wald test of parallel-lines assumption became significant (Chi-square = 31.25, p-value <0.001) indicating the need for fitting a less restrictive model, the partial proportional odds model through applying gologit2 and gsvy commands. The authors observed that using the prefix gsvy (to account for sampling stratification) in the multivariate partial proportional odds model changed the parameter estimates and level of significance compared to the baseline output of the default gologit2 auto-fit model (without gsvy prefix).

The outcome variable, QoL (Y) is categorized into three (poor, moderate and high), so the model produced two panels. The first panel contrasts category 1 (poor QoL) with category 2 (moderate QoL) and 3 (high QoL) and the second panel contrasts category 1 and 2 with category 3. An odds ratio value greater than 1 (positive coefficient) on the explanatory variable indicates that it is more likely that the respondent will be in a higher category of Y than the current one (increasing in the explanatory variable led to better levels of QoL); whereas, an odds value below 1 (negative coefficient) indicates the likelihood of being in the current or a lower category.

Since social support and satisfaction with care violated the proportional odds assumption, the odds ratios for this variable were allowed to vary between panels (AOR1 ≠ AOR2). AOR1 stands for panel one (low versus moderate or high QoL), while AOR2 refers to the second panel (low/ moderate versus high QoL). However, for the independent variables which met the parallel regression assumption (Brant test value ≥0.05), the odds ratio would be the same (AOR1 = AOR2) for the two panels.

## Ethics approval and consent to participate in the study

As this is a part of an ongoing PhD study, permission to conduct the study was obtained from the Institutional Review Board (IRB) of the College of Medicine and Health Sciences, Bahir Dar University with a protocol number 003/2021. Study participants were enrolled after giving verbal consent to participate in the study. The adequacy of oral consent was approved by the IRB and the consent was documented on participants' information sheet. Permission was obtained from the health facilities involved in the study. Moreover, confidentiality of the data obtained from the study participants and medical records have been strictly maintained through anonymizing identities and applying pertinent legal and ethical protocols.

## Results

### Characteristics of the study participants

Complete data were obtained from 1432 individuals giving rise to a response rate of 99.4%. Females constitute a slightly higher (51%) percentage in terms of sex distribution. The mean (±SD) age of the participants was 56.4 (±11.8) years. Individuals aged 45–54 years and 55–64 years accounted almost equally (27.9%) for the age distribution and those aged 65+ had a 26.9% share from the total sample (Table 1).

**Table 1. Socio-demographic characteristics of study participants attending chronic outpatient NCDs care in Bahir Dar, Ethiopia (N = 1432).**

| Variables | Frequency | Percentage |
|---|---|---|
| Age group | | |
| ≤44Yrs | 247 | 17.3 |
| 45-54Yrs | 399 | 27.9 |
| 55-64Yrs | 400 | 27.9 |
| 65+Yrs | 386 | 26.9 |
| Sex | | |
| Male | 702 | 49.0 |
| Female | 730 | 51.0 |
| Marital status | | |
| Currently married | 1081 | 75.5 |
| Single* | 351 | 24.5 |
| Education | | |
| No formal education | 780 | 54.5 |
| Primary education (Grade 1–8) | 166 | 11.6 |
| Secondary (9–12) | 171 | 11.9 |
| College level and above | 315 | 22.0 |
| Residence | | |
| Urban | 1007 | 70.3 |
| Rural | 425 | 29.7 |
| Occupation | | |
| Housewife | 329 | 23.0 |
| Employed (government and private) | 328 | 22.9 |
| Farmer | 288 | 20.1 |
| Trader | 207 | 14.5 |
| Retired | 141 | 9.8 |
| Unemployed | 139 | 9.7 |
| Wealth Index (SES) | | |
| Poorest | 269 | 18.8 |
| Poorer | 334 | 23.3 |
| Middle | 267 | 18.6 |
| Rich | 252 | 17.6 |
| Richest | 310 | 21.6 |

*Includes never married, divorced, widowed and separated

The majority of participants (75.5%) were married at the time of data collection. Looking into the education level of the respondents, a little more than half (54.5%) of them did not attend any formal education. Urban residents accounted the largest (70.3%) proportion, and housewives (23%) and employed individuals (22.9%) represent the largest proportion in the occupation category. The highest proportion of SES were those with low SES (37.4%) (Table 1).

## Body mass index (BMI)

For body mass index (BMI), the highest percentage (53.3%) were those with normal BMI, with about one third (32%) of the participants being either overweight or obese (Fig 1).

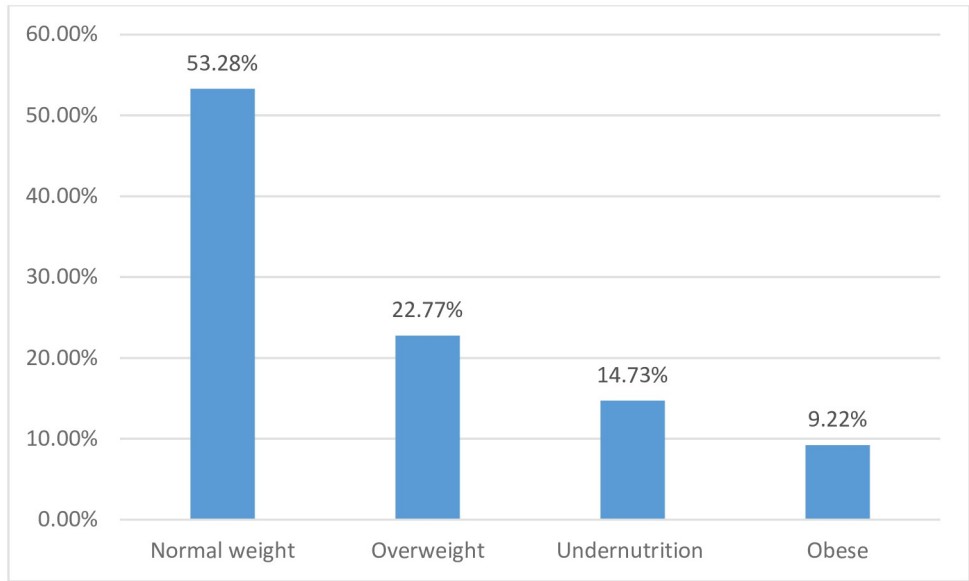

**Fig 1. Proportion of individuals in different level of nutritional status based on the BMI indices.**

## Psychosocial Characteristics

The mean of social support scale was 10.2 and a standard deviation (SD) of ± 2.17 scores. Just over half (50.7%) of the participants reported that they have moderate social support, and about one third (28%) reported strong social support, while the remaining 21% reported that they have a poor social support.

## Magnitude of NCDs and number of chronic NCDs identified per person

The magnitude of each of the chronic conditions considered in this study is shown in Fig 2. The number of NCDs identified per person ranged from one to four (mean = 1.74, SD = 0.78). Hypertension was the most frequently reported NCD (63.5%), followed by diabetes (42.5%) and heart diseases (25.6%).

## Magnitude of multimorbidity

More than half 54.8% (CI = 52.2%, 57.4%) of the study participants had multimorbidity, with 39.6% having two chronic NCDs and 15.2% three or more chronic NCDs (Fig 3).

The most prevalent NCDs greatly contributed to shaping the patterns of multimorbidity in this study. For example, hypertension co-existed with diabetes and heart diseases in 38.2% and 19.0% of the participants, respectively. Similarly, co-occurrence of diabetes was observed among individuals with heart diseases, depression and other types of reported chronic conditions. Hypertension remained the most frequently reported NCD (87.2%) among individuals living with three or more NCDs in our study. Diabetes was reported by 51% of those who had three or more chronic NCDs and heart diseases were reported by 39% of the participants from this group (Table 2).

A third (33.5%) of the study participants had poor quality of life and about a quarter of them had moderate QoL (Fig 4).

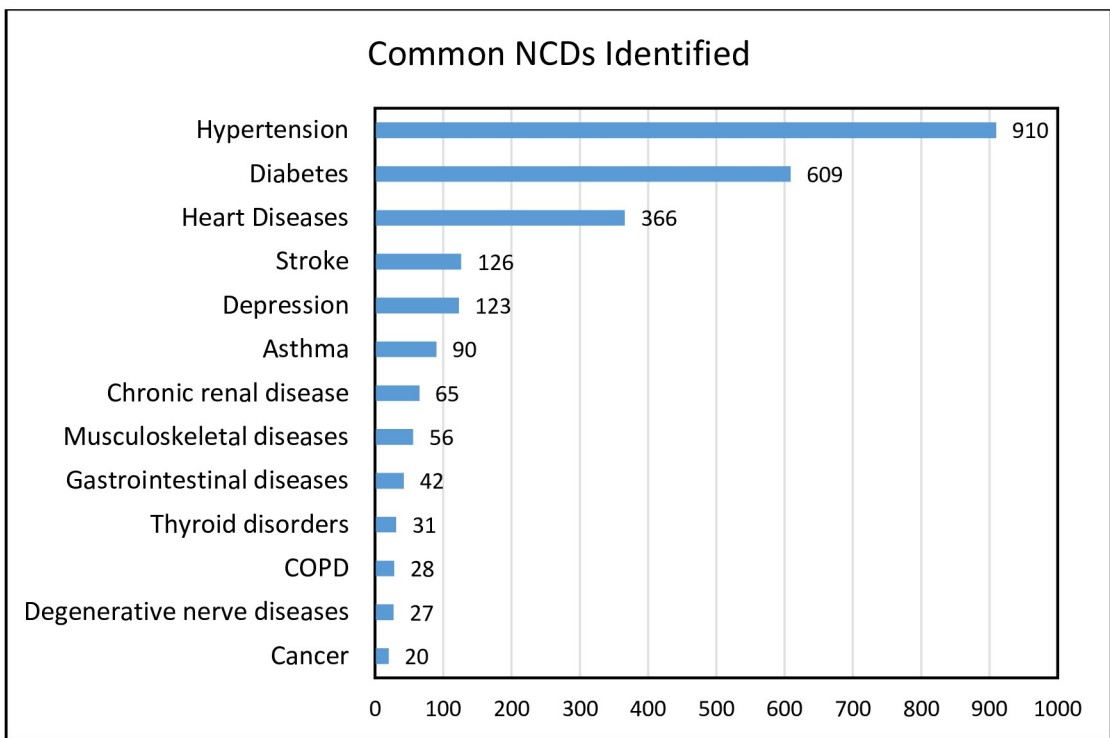

**Fig 2. List of NCDs studied and their magnitude among participants attending chronic outpatient NCDs care, Bahir Dar, Ethiopia (N = 1432).**

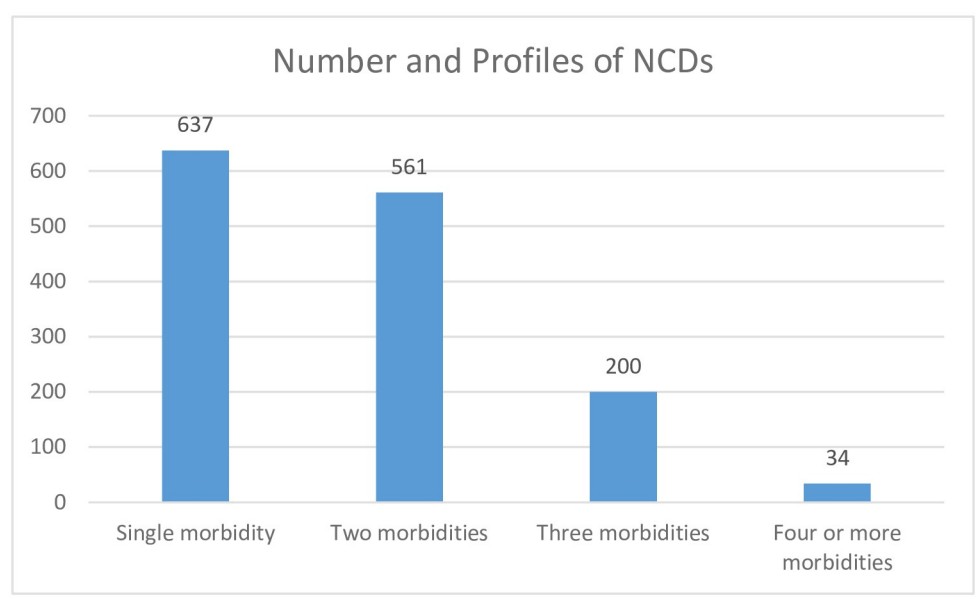

**Fig 3. Patterns of NCDs morbidity among individuals attending chronic NCDs care in Bahir Dar, Ethiopia (N = 1432).**

**Table 2. Distribution of individual NCDs and their pairwise and triple or quadruple combination, among people attending chronic outpatient NCD care in Bahir Dar, Ethiopia (N = 1432).**

| Single morbidity | | Common pairs of NCDS | | Common Triples of NCDs | |
|---|---|---|---|---|---|
| Chronic NCD | Frequency (%) | Combination | Frequency (%) | Combination | Frequency (%) |
| Hypertension alone | 245 (37.9) | Hypertension +Diabetes | 217 (38.2) | Hypertension +Diabetes+ heart diseases | 19 (8.7) |
| Diabetes alone | 225 (34.8) | Hypertension + Heart diseases | 108 (19.0) | Hypertension +Diabetes + depression | 18 (8.3) |
| Heart diseases alone | 120 (18.5) | Hypertension + stroke | 38 (6.7) | Hypertension +Diabetes + other NCDs | 49 (22.5) |
| All other forms of single NCDs[a] | 57 (8.8) | Hypertension +Musculoskeletal diseases | 23 (4.0) | Hypertension +heart diseases + other NCDs | 43 (19.7) |
| | | Hypertension + Asthma | 21 (3.7) | Hypertension + Diabetes + heart diseases + other NCDs | 12 (5.5) |
| | | Hypertension + Chronic Renal diseases | 21 (3.7) | Hypertension + Diabetes + two other NCDs | 13 (6.0) |
| | | Hypertension + Depression | 18 (3.2) | Hypertension + two or other NCDs | 36 (16.5) |
| | | Hypertension + other chronic diseases | 25 (4.4) | Diabetes + two or more other NCDs | 13 (6.0) |
| | | Diabetes + Depression | 8 (1.4) | Heart diseases + two or more other NCDs | 11 (5.0) |
| | | Diabetes + heart disease | 6 (1.0) | Triple or quadruple of all other NCDs | 4 (1.8) |
| | | Diabetes + other chronic NCDs | 25 (4.4) | | |
| | | Heart disease +Depression | 16 (2.8) | | |
| | | Heart diseases + other chronic diseases | 27 (4.8) | | |
| | | Comorbidity of other NCDs | 14 (2.5) | | |

[a] Includes asthma, COPD, stroke, cancer and depression.

Other NCDs include degenerative nerve diseases, thyroid disease and gastrointestinal diseases.

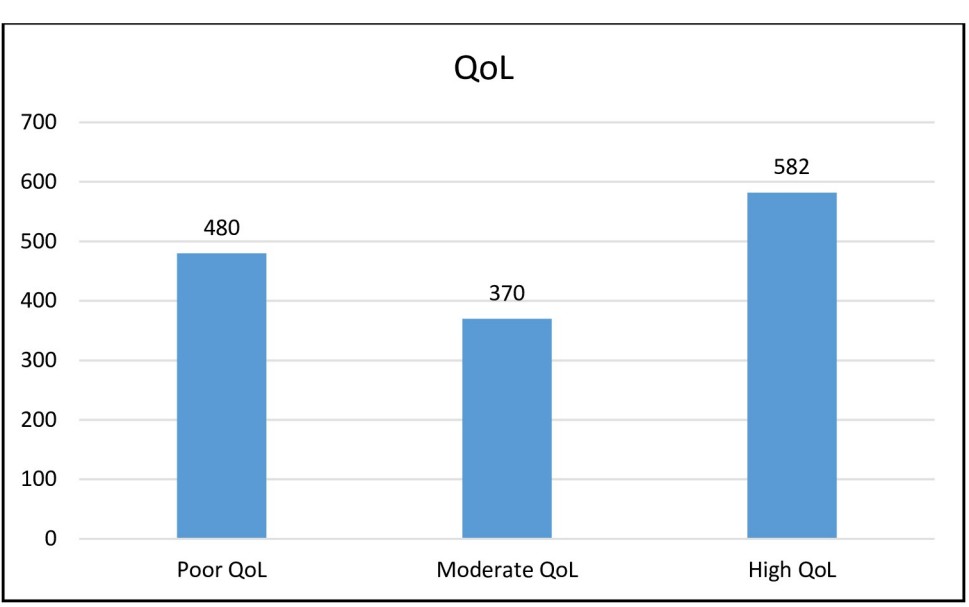

**Fig 4. Number of individuals classified in different categories of health-related QoL.**

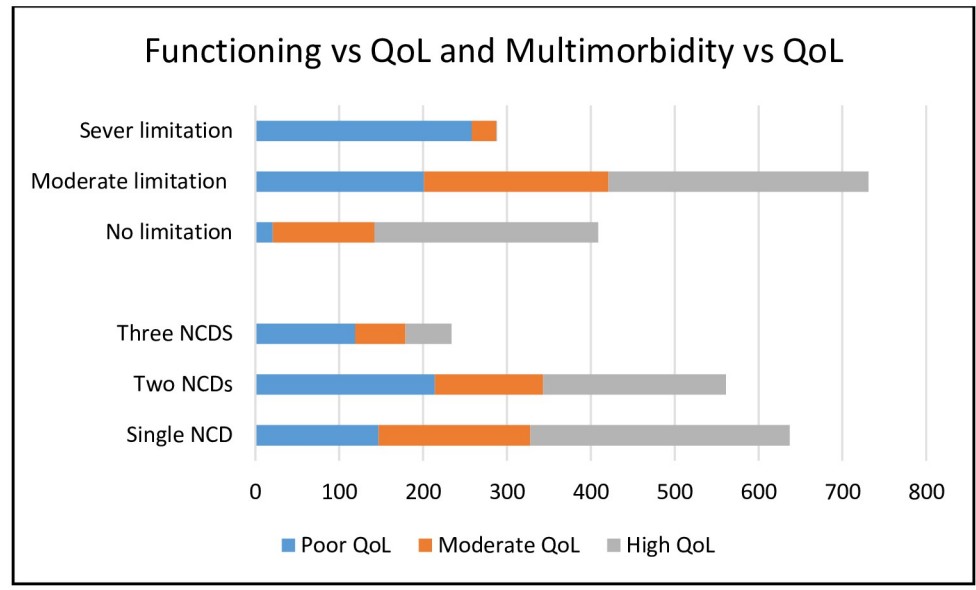

**Fig 5. Graphic presentation of the relationship between QoL, functioning and multimorbidity.**

Individuals with multimorbidity had a relatively poor QoL than those without multimorbidity (62% vs. 38%). Similarly, a higher proportion of individuals with severe functional limitations had poor QoL compared to those without severe limitations (Fig 5).

Table 3 compares the output between the default univariate ordinal logistic regression analysis (ologit) model and the model adjusted for stratification (using the svy: ologit command). Although there are difference in the parameter estimates between the two models, age, BMI, functioning, satisfaction with care and presence of multimorbidity remain significantly associated with QoL in the model accounting for stratification. However; sex, SES and perceived social support lost their significance in the model accounting for a stratified sampling (Table 3).

## Multivariable partial proportional odds analysis

In the final model, we entered all the variables that had a P-value <0.2 in the univariate model adjusted for stratification. As indicated above, social support score and satisfaction with care did not satisfy the parallel lines regression assumptions, necessitating a fitting of the partial proportional odds (PPO) model. Therefore, for the two variables, the odds ratios were allowed to vary (AOR1 ≠ AOR2) in the two panels.

Whereas age, sex, SES, BMI, multimorbidity and perceived functional status met the parallel regression assumption (Brant test value ≥0.05). Hence, the odds ratio would be the same (AOR1 = AOR2) for the two panels. In the final model, AOR1 stands for panel one (low versus moderate or high QoL), while AOR2 refers to the second panel (low/ moderate versus high QoL).

Looking into the final model (Table 4), statistically significant differences were observed in terms of the effect of most of the explanatory variables on QoL, adjusting for all the covariates. However, self-reported functioning lost its significance in the final model (specified by gsvy: gologit2, Y x1, x2, x3. . . auto lrforce command). While social support score became significant in the final model (Table 4).

**Table 3. Univariate ordinal logistic regression analysis (default and using svy prefix) and Brant test estimates for the proportional odds assumptions.**

| Variables | QoL category | | | P-value | P-value: svy output(accounting for stratification) | Brant test |
|---|---|---|---|---|---|---|
| | Poor QoL | Moderate QoL | High QoL | | | |
| Residence | | | | | | |
| Urban | 272 | 273 | 462 | base | | |
| Rural | 208 | 97 | 120 | 0.474 | | |
| Sex | | | | | | |
| Male | 207 | 174 | 321 | base | base | <0.564 |
| Female | 273 | 196 | 262 | <0.0.001 | 0.072 | |
| Age | | | | | | |
| Mean | 60.0 | 56.0 | 53.6 | <0.001 | 0.037* | 0.377 |
| SD | 12.61 | 11.30 | 10.62 | | | |
| Education | | | | | | |
| Below primary | 360 | 191 | 229 | 0.557 | | |
| Primary and above | 120 | 179 | 353 | base | | |
| Marital status | | | | | | |
| Married | 312 | 284 | 486 | base | | |
| Single | 168 | 87 | 96 | <0.648 | | |
| BMI | | | | | | |
| ≤24.99 | 250 | 209 | 304 | base | base | 0.665 |
| ≥25 | 115 | 41 | 55 | <0.023 | 0.020* | |
| SES | | | | | | |
| Low | 234 | 128 | 174 | base | base | <0.365 |
| Middle or high | 133 | 109 | 163 | 0.003 | 0.173 | |
| Social supportscale | | | | | | |
| Mean | 9.3 | 10.6 | 10.7 | <0.001 | 0.092 | <0.001** |
| SD | 2.17 | 1.70 | 2.19 | | | |
| Overall functioning | | | | | | |
| Limited/weak capacity | 258 | 29 | 5 | base | base | <0.140 |
| Strong capacity | 222 | 341 | 577 | <0.001 | 0.006* | |
| Care satisfaction | | | | | | |
| not satisfied | 108 | 43 | 32 | base | base | <0.001** |
| Satisfied | 372 | 327 | 550 | 0.019* | 0.018* | |
| Multimorbidity | | | | | | |
| No | 182 | 170 | 295 | base | Base | 0.478 |
| Yes | 298 | 200 | 287 | <0.001* | 0.011* | |

*Statistically significant at p-value 0.05 using svy: ologit command,

**variables that violated the PPO assumptions

The final model indicates that the odds of being in the combined categories of moderate and high QoL versus poor QoL was 2.6 times higher for patients satisfied with care than individuals that are not satisfied, when holding the other variables constant [AOR1 = 2.6 (95% CI: 1.83, 5.37)]. However, the odds of being in the combined categories of moderate and low versus high QoL was lower by a factor of 0.07 for satisfied patients than patients which were not satisfied. It was not, however, statistically significant [AOR2 = 0.93(95% CI: 0.57, 1.51)].

In this study, for a unit increase in the social support scale, the odds of being in the higher categories of QoL versus lower categories was 1.41 times greater [AOR1 = 1.41(95% CI: 1.29, 1.56)], given the other variables held constant.

**Table 4.  Multivariable partial proportional odds model showing the association between factors and QoL.**

| Explanatory Variables | Outcome variables (panels) | | | | |
|---|---|---|---|---|---|
| | Panel One (1 Vs. 2 and 3) | | | Panel Two (1 or 2 Vs 3) | |
| | AOR 1 (95%CI) | Coefficients constant (OR1 = OR2) | P-value | AOR2(95%CI) | P-value |
| Sex (Female vs. male[Ref]) | | 0.63(0.27, 1.46) | 0.141 | | |
| SES (high vs. low [ref]) | | 0.73(0.41, 1.29) | 0.138 | | |
| Self-rated functioning (Strong vs. weak [Ref]) | | 1.28(0.66, 3.16) | 0.233 | | |
| Satisfaction (satisfied vs. not satisfied [Ref]) | 0.93(0.57, 1.51) | | 0.588 | **2.63(1.83, 5.37)** | **0.023** |
| Social support scale | **1.41(1.29, 1.56)** | | **0.004** | 1.15(0.88,, 1.50) | <0.153 |
| Multimorbidity (yes vs.no [Ref]) | | **0.76(0.69,, 0.84)** | **0.008** | | |
| Age in years | | **0.96(0.94, 0.99)** | **<0.032** | | |
| BMI ($\geq$ 25 vs. <24.99 [Ref]) | | **0.80(0.66, 0.98)** | **0.043** | | |

Meanwhile, the variables that did not violate the PPO assumption had a constant beta coefficient (AOR1 = AOR2) for each of the two QoL categories; hence a single odds ratio was reported. We found that, for every one-year increase in age of the participants, the odds of being in the poorer QoL category was increased by a factor of 0.04 [AOR1 = AOR2: 0. 96 (95% CI: 0.94, 0.99)].

Similarly, the odds of being in the higher categories of QoL was 0.24 times lower for individuals with multimorbidity than those people without multimorbidity [AOR1 = AOR2: 0.76 (95% CI: 0.69, 0.84)]. Moreover, individuals having a higher categories of BMI score were 20% more likely to have a poorer QoL than individuals in the lower BMI category, holding the other variable remain constant [AOR1 = AOR2: 0. 80 (95% CI: 0.66, 0.98)] (Table 4).

## Discussion

Understanding the effect of multimorbidity on health related quality of life (QoL) is one of the top research priorities in the existing literature [20, 81]. A broad sample of health facilities where most of the people living with chronic NCDs receive their care and corresponding number of patients were randomly selected and enrolled to determine the magnitude of multimorbidity and its association with QoL in the study area. We employed a blend of methods (face-to-face interview and review of medical records) to better determine the presence of individual NCDs and their pairwise and triple combination among a broad sample of 1432 individuals (aged 40+) attending chronic medical care in hospitals and specialized health facilities in Bahir Dar city, Northwest Ethiopia.

The implication of our findings should be interpreted in light of the variations in the way QoL has been measured and analyzed globally. The method of analysis we employed- the partial proportional odds (PPO) is relatively new in the context of analyzing QoL data [53]. Compared to other methods, the PPO model is said to be a robust analytic method for categorical QoL data [48, 53].

In this study, the authors found that multimorbidity is common, affecting the majority (55%) of the individuals receiving chronic outpatient medical care. The high burden of multimorbidity in the study area implies that individuals living with chronic conditions have already been facing the overwhelming consequences of multimorbidity. Previous studies have also reported the challenges of living with multimorbidity in the country [8]. Studies show the most common risk factors contributing for the increasing burden of multimorbidity are advanced age, obesity, physical inactivity, socioeconomic deprivation and use of tobacco and alcohol [21]. This implies, the majority of the risk factors for multimorbidity are modifiable [6].

It was found that a third of (33.5%) of individuals living with chronic NCDs had poor QoL, with 62% of these having multimorbidity. Several studies have shown that multimorbidity is a key factor contributing for poor QoL [38, 39, 43]. Although direct comparison may not be possible with most of the previous studies owing to methodological variations, the authors observed that patients with multimorbidity had significantly poorer quality of life compared to patients with single chronic conditions. Studies which utilized the PPO model to analyze QoL data have also reported consistent results corroborating the negative association between multimorbidity and QoL [53]. However, evidence shows that not only the mere sum of individuals conditions, but also the nature of disease clusters matter in relation to quality of life, functionality and survival [64].

Consistent with previous studies [17, 24], people with advanced age were shown to have reduced QoL in our study. Advanced age is known to impair molecular and cellular functions that leads to a gradual decline in the physiological reserves and capacity of the individuals [82]. The observed inverse relationship between advanced age and poor QoL may also be due to the mediated effect of multimorbidity as the probability of having multimorbidity was higher among the middle-aged and elderly in our study. Although it is expected that people in old age often face poor QoL due to physical disability, frailty and sensory impairment [29], earlier onset of multimorbidity and its effect on QoL was reported to be higher among young adults living in socioeconomically deprived areas [40].

Evidence shows a clear relationship between obesity and poor QoL. In this study, individuals having higher score of BMI had lower levels of QoL than their leaner counterparts. However, the association between higher scores of BMI and poor QoL might be due to the mediating effect of multimorbidity. It was observed in this study that obese individuals had higher odds of multimorbidity. On the other hand, individual having higher BMI scores could have increased risk of physical limitations [25, 83], which will contribute negatively to QoL [84]. The complex interplay between obesity, multimorbidity and poor QoL has also been well established globally [2].

Medical care alone cannot adequately improve QoL [25]. The presence of strong social support is helpful to improve patient's adaptation to life and their QoL [85]. The authors found a positive and statistically significant association between perceived social support and QoL. However, some studies were inconsistent in reporting the effect of social support in modifying QoL among individuals with multimorbidity [86–88]. People with multimorbidity are generally less satisfied with the care they receive [2]. Ensuring satisfaction with care for people with multiple chronic health conditions is challenging because the notion of satisfaction is influenced by several actors, including caregivers, healthcare providers and the health system in general [25]. In this study, individuals satisfied with care were more likely than their counterparts to have higher odds of better QoL. This is consistent with previous findings [89]. However, other literature shows no difference between satisfaction with care and multimorbidity related QoL [35, 90].

Previous studies have shown that economically deprived people struggle to cope with everyday life activities and have a lower quality of life compared with more affluent patients with multimorbidity [15]. Further, multimorbidity was associated with a more significant reductions in QoL scores amongst participants living in the most deprived areas [40], signifying a coupling effect of poverty and multimorbidity on QoL. However, the authors did not find any statistically significant relationship between SES and QoL. This might be due to the difference in methodology or nature of study participants involved in our study.

## Implication for practice and research

The main goal of health care for the people living multiple chronic conditions is to help them achieve better QoL [25, 91]. Given that the magnitude of multimorbidity is high and that it

poses a profound impact on QoL, the healthcare system needs to be guided by these findings in order to adequately respond to individual patient needs. Care for people with multimorbidity must be based on the needs and circumstances of the person as a whole rather than the different conditions a person happens to have [90]. The provision of patient-centered care in which all healthcare providers work together with patients to ensure coordination, consistency and continuity of care over time is essential [92]. This will in turn improve the wellbeing and survival of the people with multimorbidity in the study area.

The evidence base on the association between multimorbidity and QoL is growing, albeit slowly. However, the methodologies employed to study multimorbidity vary widely [3], and the methods applied to investigate the impact of multimorbidity on QoL have not been universally consistent [40]. We are aware of the possible limitation of comparing our results with studies that employed different tools and methods of analysis. Further research is needed on the application of ordinal regression and PPO models for analyzing QoL data and to identify the associated covariates. Understanding the longitudinal effect of individual NCDs, multimorbidity and disease severity on QoL would help fill the substantial gaps in our knowledge in this regard. It is also imperative to study the way health systems are organized to manage patients with multimorbidity, and to explore the perspective and lived experiences of individuals with multiple chronic conditions in Ethiopia.

### Strength and limitations of the study

Our study has the advantage of involving a broader range of health facilities rendering comprehensive care for the people with chronic NCDs. Guided by a published study protocol; this study employed three complementary methods to define the presence of chronic NCDs accurately. The authors utilized the widely accepted QoL measure, the SF-12V2 tool and analyzed the data using a relatively robust categorical data analytic method, the PPO model. The PPO model we fitted has taken into account variation in stratified sampling to analyze the QoL data and identify covariates associated with QoL in a relatively efficient, reliable and valid way. However, the findings of this facility-based study may not exactly represent the underlying epidemiology of multimorbidity and patterns of association between multimorbidity and QoL in the general population in region and beyond. It is also difficult to confirm that the observed association between the variables has a temporal relationship. Further, variables measured by Likert-type scales in general are subject to bias; therefore, care should be taken when interpreting studies using such scales. In addition, the lack of consistent methods to measure both multimorbidity and QoL globally makes our findings comparable to only some of the previous studies. However, the application of the PPO model makes our study more parsimonious than studies employing the traditional linear methods of analyzing QoL data in the field of multimorbidity research in the global context.

**Conclusion and Recommendations.** The magnitude of multimorbidity in this study was high and the highest proportion of individuals with multimorbidity had poor QoL. The high multimorbidity estimate observed might be attributed to the fact that the study was conducted among health facilities where most of people living with chronic NCDs were attending care. Advanced age, living with multimorbidity and obesity were the variables negatively associated with QoL. In contrast, high-perceived social support and satisfaction with care were the variables associated with higher categories of QoL.

The literature on the relationship between multimorbidity and QoL is dominated by studies in high income countries. If health systems in LMICs are to meet the needs of the people with multimorbidity, it is essential to understand the full breadth of multimorbidity across the ages and its effect on individuals QoL, functioning and survival. Future studies may need to focus

on understanding the epidemiology of multimorbidity and its effect on QoL and survival in the general population. Further studies areneeded to explore the longitudinal effect of multimorbidity on quality of life, functioning and survival, and to assess the way health services organized to meet the care needs of the people with multiple chronic conditions in the country. It is also imperative to replicate the methods that were employed to measure and analyze QoL data in this study in order to facilitate comparison and further development of the approaches.

## Patient and public involvement

No patient or the public were involved in the design, or conduct, or reporting, or dissemination plans of our research

## Supporting information

**S1 Data. Data used for the QoL study and multimorbidity.**
(SAV)

**S1 Output. Output of PPO model using Svy do file August 30 (PGPH).**
(DO)

**S1 File. Short Form (SF12) QoL assessment tool.**
(PDF)

## Acknowledgments

We also would like to thank data collectors, supervisors, facilities leaders and study participants for their support in making this study a reality.

Fantu Abebe extends his gratitude to Jhpiego-Ethiopia for the facilities he used while conducting this study.

## Author Contributions

**Conceptualization:** Fantu Abebe Eyowas, Marguerite Schneider, Shitaye Alemu Balcha, Sanghamitra Pati, Fentie Ambaw Getahun.

**Data curation:** Fantu Abebe Eyowas, Fentie Ambaw Getahun.

**Formal analysis:** Fantu Abebe Eyowas.

**Funding acquisition:** Fantu Abebe Eyowas.

**Investigation:** Fantu Abebe Eyowas, Fentie Ambaw Getahun.

**Methodology:** Fantu Abebe Eyowas, Marguerite Schneider, Shitaye Alemu Balcha, Fentie Ambaw Getahun.

**Project administration:** Fantu Abebe Eyowas, Fentie Ambaw Getahun.

**Resources:** Fantu Abebe Eyowas.

**Software:** Fantu Abebe Eyowas.

**Supervision:** Marguerite Schneider, Shitaye Alemu Balcha, Sanghamitra Pati, Fentie Ambaw Getahun.

**Validation:** Fantu Abebe Eyowas, Marguerite Schneider, Shitaye Alemu Balcha, Sanghamitra Pati, Fentie Ambaw Getahun.

**Writing – original draft:** Fantu Abebe Eyowas.

**Writing – review & editing:** Marguerite Schneider, Shitaye Alemu Balcha, Sanghamitra Pati, Fentie Ambaw Getahun.

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
