## [Decision Letter · Decision Letter 0]

29 Aug 2022

PGPH-D-22-01200

Multimorbidity and health-related quality of life among patients attending chronic outpatient medical care in Bahir Dar, Northwest Ethiopia: the application of partial proportional odds model

Dear Dr. Eyowas,

Thank you for submitting your manuscript to PLOS Global Public Health. After careful consideration, we feel that it has merit but does not fully meet PLOS Global Public Health’s publication criteria as it currently stands. Therefore, we invite you to submit a revised version of the manuscript that addresses the points raised during the review process.

EDITOR: Please address the critical comments of the two reviewers. It's important that you re-check for any grammatical and punctuation problems throughout the manuscript. 

We look forward to receiving your revised manuscript.

Kind regards,

Tilahun Haregu

Academic Editor

Journal Requirements:

1. Please update your online Competing Interests statement. If you have no competing interests to declare, please state: “The authors have declared that no competing interests exist.”

2. Please amend your online detailed Financial Disclosure statement. This is published with the article. It must therefore be completed in full sentences and contain the exact wording you wish to be published.

3. Please provide a complete Data Availability Statement in the submission form, ensuring you include all necessary access information or a reason for why you are unable to make your data freely accessible. If your research concerns only data provided within your submission, please write “All data are in the manuscript and/or supporting information files.” as your Data Availability Statement.

4. Please include a title page at the beginning of your manuscript file, that lists full author names and institute addresses. Do not upload as a separate file.

5. Your manuscript is missing the following sections: Abstract. Please ensure these are present, and in the correct order, and that any references to subheadings in your main text are correct. An outline of the required sections can be consulted in our submission guidelines here: https://journals.plos.org/globalpublichealth/s/submission-guidelines#loc-parts-of-a-submission

6. Please provide separate figure files in .tif or .eps format only and ensure that all files are under our size limit of 10MB.

7. We have noticed that you have uploaded Supporting Information files, but you have not included a list of legends. Please add a full list of legends for your Supporting Information files after the references list.

Additional Editor Comments (if provided):

Reviewers' comments:

Reviewer's Responses to Questions

**Comments to the Author**

1. Does this manuscript meet PLOS Global Public Health’s publication criteria? Is the manuscript technically sound, and do the data support the conclusions? The manuscript must describe methodologically and ethically rigorous research with conclusions that are appropriately drawn based on the data presented.

Reviewer #1: Yes

Reviewer #2: Yes

2. Has the statistical analysis been performed appropriately and rigorously?

Reviewer #1: No

Reviewer #2: Yes

3. Have the authors made all data underlying the findings in their manuscript fully available (please refer to the Data Availability Statement at the start of the manuscript PDF file)?

Reviewer #1: Yes

Reviewer #2: Yes

4. Is the manuscript presented in an intelligible fashion and written in standard English?

Reviewer #1: Yes

Reviewer #2: Yes

5. Review Comments to the Author

Reviewer #1: This is a well written article. The study is well-conceived (I read your published protocol) and very worthwhile.

I liked everything very much except that I would like you to please consider redoing the univariate and multivariate regression. These should be done using Stata's survey module. As you know, I am sure, first svyset. Then svy: for the univariate and gsvy: for the gologit2 command.

I also feel that the structure of the sampling should be a bit clearer in the methods section. At the moment you mention stratification but the strata appear to be more than two and this look more like PSUs to me; whereas, after reading your protocol it is clear there were two strata; public and private sectors. Then there was two stage random and systematic sampling within these two strata. Normally one tries to meet the sample size for each stratum. In your case you allocated numbers pro rata to each based on historical utilisation. This may make the weighting easier to deal with. Nevertheless the two Strata should be identified as well as the clusters within each stratum. It would also be best practice to publish a table of design effects so that other researchers can use these when deciding on their own sample sizes.

Reviewer #2: This is a great study that adds to the NCD literature of multimorbidity in LMIC. Data was collected through two different sources which strengthens the study data quality.

Introduction:

The introduction is very thorough and complete however it would benefit from additional reviews or studies done in LMIC or regional/local (Ethiopia) studies for multimorbidity:

• Bhagavathula AS,. Prevalence and Determinants of Multimorbidity, Polypharmacy, and Potentially Inappropriate Medication Use in the Older Outpatients: Findings from EuroAgeism H2020 ESR7 Project in Ethiopia. Pharmaceuticals (Basel). 2021 Aug 25;14(9):844. doi: 10.3390/ph14090844. PMID: 34577544; PMCID: PMC8468438.

• Abebe F, Schneider M, Asrat B, Ambaw F. Multimorbidity of chronic non-communicable diseases in low- and middle-income countries: A scoping review. J Comorb. 2020 Oct 16;10:2235042X20961919. doi: 10.1177/2235042X20961919. PMID: 33117722; PMCID: PMC7573723.

Methods:

The study methodology has been clearly described.

The authors mention the use of SF-12 as a data collection instrument. Authors should consider to add the SF-12 tool as an appendix

Results

Line 236 Lifestyle and psychosocial characteristics: It is confusing that the authors talk about BMI and include Figure 1, but in the same paragraph they talk about social support – which there is no graph or table for it. If left as is I would recommend splitting the paragraph and adding a note saying (data not shown in tables)

Line 275 -314 would fit better under methods rather than results

Discussion

It would be interesting to include risk factors into the study as well (tobacco use, physical inactivity, obesity, alcohol use) – this could be added as a discussion point. Freisling H, et,.al. Lifestyle factors and risk of multimorbidity of cancer and cardiometabolic diseases: a multinational cohort study. BMC Med. 2020 Jan 10;18(1):5. doi: 10.1186/s12916-019-1474-7. PMID: 31918762; PMCID: PMC6953215.

The following statement made earlier in the discussion should go under study strengths and limitations:

The authors used the very commonly used QoL measure, the SF-12V2 362 tool. However, the method of analysis we employed- the partial proportional odds (PPO) is relatively new 22 363 in the context of analyzing QoL data(77). The PPO model is said to be a robust QoL data analytic method 364 compared to other methods of analysis provided the nature of a given data warrants its use(41).

Table 2

Superscript a mentions the definition of all other NCDs which the footnote specifies to be asthma, COPD, stroke, cancer and depression. However in the methods (and in Figure 2) the authors mention many more NCDs not mentioned in Table 2. Please clarify for the reader if these other chronic conditions were taken out of the analysis (degenerative nerve diseases, thyroid diseases, gastrointestinal diseases)

Figures:

Figure 1- strongly recommend not using pie charts for visualizing more than 2 groups: The issue with pie chart (data-to-viz.com)

Figures already have at the bottom the legend and description so recommend deleting the top title to avoid confusion (for example: Figure 2 Common NCDs identified Vs List of NCDS studied and their magnitude …)

Please add what the numbers in the graphs represent (percentages, N, etc)

Figure 5 – check for a mistake/typo in “severe”.

6. PLOS authors have the option to publish the peer review history of their article (what does this mean?). If published, this will include your full peer review and any attached files.

**Do you want your identity to be public for this peer review?** For information about this choice, including consent withdrawal, please see our Privacy Policy.

Reviewer #1: No

Reviewer #2: No

---

## [Editor Report · Decision Letter 1]

3 Oct 2022

PGPH-D-22-01200R1

Multimorbidity and health-related quality of life among patients attending chronic outpatient medical care in Bahir Dar, Northwest Ethiopia: the application of partial proportional odds model

Dear Dr. Eyowas,

Thank you for submitting your manuscript to PLOS Global Public Health. After careful consideration, we feel that it has merit but does not fully meet PLOS Global Public Health’s publication criteria as it currently stands. Therefore, we invite you to submit a revised version of the manuscript that addresses the points raised during the review process.

Thanks for addressing the comments raised by the reviewer. The paper would benefit from the use of an appropriate multimorbidity index (see:https://www.bmj.com/content/368/bmj.m160)

We look forward to receiving your revised manuscript.

Kind regards,

Tilahun Haregu

Academic Editor

Journal Requirements:

Additional Editor Comments (if provided):

Thanks for addressing the comments raised by the reviewer. The paper would benefit from the use of an appropriate multimorbidity index (see:https://www.bmj.com/content/368/bmj.m160)
---

## [Editor Report · Decision Letter 2]

5 Oct 2022

Multimorbidity and health-related quality of life among patients attending chronic outpatient medical care in Bahir Dar, Northwest Ethiopia: the application of partial proportional odds model

PGPH-D-22-01200R2

Dear Mr. Eyowas,

We are pleased to inform you that your manuscript 'Multimorbidity and health-related quality of life among patients attending chronic outpatient medical care in Bahir Dar, Northwest Ethiopia: the application of partial proportional odds model' has been provisionally accepted for publication in PLOS Global Public Health.

Best regards,

Tilahun Haregu

Academic Editor